# Mechanotransduction in Wound Healing and Fibrosis

**DOI:** 10.3390/jcm9051423

**Published:** 2020-05-11

**Authors:** Britta Kuehlmann, Clark A. Bonham, Isabel Zucal, Lukas Prantl, Geoffrey C. Gurtner

**Affiliations:** 1Division of Plastic and Reconstructive Surgery, Department of Surgery, Stanford University, Stanford, CA 94305, USA; bkuehl@stanford.edu (B.K.); cbonham@stanford.edu (C.A.B.); 2University Center for Plastic, Reconstructive, Aesthetic and Hand Surgery, University Hospital Regensburg and Caritas Hospital St. Josef, 93053 Regensburg, Germany; isabel.zucal@icloud.com (I.Z.); Lukas.Prantl@klinik.uni-regensburg.de (L.P.)

**Keywords:** mechanotransduction, fibrosis, wound healing

## Abstract

Skin injury is a common occurrence and mechanical forces are known to significantly impact the biological processes of skin regeneration and wound healing. Immediately following the disruption of the skin, the process of wound healing begins, bringing together numerous cell types to collaborate in several sequential phases. These cells produce a multitude of molecules and initiate multiple signaling pathways that are associated with skin disorders and abnormal wound healing, including hypertrophic scars, keloids, and chronic wounds. Studies have shown that mechanical forces can alter the microenvironment of a healing wound, causing changes in cellular function, motility, and signaling. A better understanding of the mechanobiology of cells in the skin is essential in the development of efficacious therapeutics to reduce skin disorders, normalize abnormal wound healing, and minimize scar formation.

## 1. Introduction

Wound healing is a complex process of overlapping, consecutive phases that restores the structural integrity of the skin following injury. Recent studies have identified mechanical signaling networks in the skin that influence wound healing. These pathways elicit changes in the physiology and structure of the skin, which can delay regeneration and lead to fibrosis. In recent years, the effects of force and mechanical stress on wound healing have gained significant clinical attention and the importance of these pathways has been confirmed in clinical trials. Cells have been shown to respond to mechanical stress by altering their functional, migratory, and signaling capabilities through a process called mechanotransduction. All such events have a profound effect on wound healing, leading to changes in the final wound phenotype, including overhealing (fibrosis, keloids) and underhealing (chronic wounds).

## 2. Wound Healing and Fibrosis in the Skin

Wound healing undergoes several distinct phases following injury to the skin: hemostasis, inflammation, proliferation, and tissue remodeling. Immediately after tissue disruption, platelets converge and adhere to severed blood vessels and prevent excessive bleeding. These fibrin clots provide a variety of beneficial proteins and signaling molecules to progress healing into the inflammatory phase [1].

Circulating monocytes undergo chemotaxis as a result of inflammatory cell signaling, migrating to the wound bed where other signaling molecules induce their differentiation into macrophages. Along with newly recruited neutrophils and tissue-resident macrophages, these migratory macrophages attempt to clean the wound bed of harmful and foreign substances. In chronic wounds, this phase is drawn out, and it often does not come to a successful conclusion, inhibiting the proper, successful progression of wound healing. 

In acute wounds, following the successful cellular debridement of the wound, macrophages and other cells begin to secrete signaling molecules in order to recruit fibroblasts to the wound. These fibroblasts undergo differentiation into myofibroblasts, characterized by their alpha smooth muscle actin (α-SMA) bundles that give them a contractile capability [2]. These myofibroblasts synthesize collagen and other extracellular matrix (ECM) components that serve as the foundation for the healing wound [3]. The myofibroblasts work to contract the wound following sufficient collagen and ECM deposition. 

As the wound begins to close, cell signaling causes phenotypical changes within the cells and the focus shifts to restructuring the newly deposited components to produce the final healed scar. Matrix metalloproteinases (MMPs) and their respective inhibitors, tissue inhibitors of metalloproteinases (TIMPs), are produced to break down and remodel the collagen and ECM bundles [4]. This final phase concludes wound healing but yields a scar that differs in structure and tensile strength when compared to regular, healthy skin. 

Scarring can differ among individuals, with some being more susceptible to keloids and hypertrophic scars than others. Most scars cause distress to the individual and they can lead to functional deficiencies, with billions of dollars being spent on scar treatment each year [5]. Thus, a better understanding of scar formation might lead to greater prevention of undesirable outcomes.

## 3. Mechanotransduction in Skin and Wounds 

Human skin constantly deals with intrinsic and extrinsic forces throughout life. The effect of mechanical force is dependent upon the stiffness and biomechanical properties of the skin, which vary between anatomical locations [6]. Mechanical stimuli contribute to alterations in the wound healing process and underlie the increased susceptibility to excessive scar formation found in particular regions of the body [7,8]. Modern plastic surgery already employs several mechanomodulatory procedures in order to counter these effects, including z-plasty and the use of steristrips [9,10]. Both relax skin tension at the site of the wound, effectively reducing scar development, although there remains room for improvement [11,12,13]. As our understanding of mechanotransduction grows, we are focusing on “next generation” biomolecular approaches to further reduce scarring and fibrosis. 

It is important to first grasp how mechanotransduction works at the cellular and tissue levels to better understand how the signaling pathways involved in wound healing and skin fibrosis are affected by mechanical force. As physical force is applied to the skin, mechanical signals are conveyed into chemical information by molecules that transmit this information to the cell (model of cellular tensegrity) [14,15,16,17]. The extracellular matrix (ECM) and the extracellular fluid (ECF) are essential to the transduction of mechanical forces into cells and control the constant remodeling of the skin. Transmembrane structures play an integral role in this process, as the membrane itself is fluid and requires specialized structures to transduce forces. Components of the cell membrane and cytoskeleton (e.g., actin and RhoA), ion channels, catenin complexes, cell adhesion molecules (e.g., focal adhesions and integrins), and several signaling pathways (e.g., Wnt, FAK-ERK, MAPK/ERK) are known to act as mechanosensors [18,19,20]. These sensors transmit mechanical signals to cells that bind to the extracellular matrix (ECM) and trigger a further signaling cascade of responses [21,22,23] (Figure 1). For instance, the destruction of the ECM in aged skin leads to the further breakdown of fibroblasts within the dermis, as these cells thereby cease to receive mechanical information [24]. As a result, collagen production is decreased, while that of collagen-degrading enzymes, such as matrix metalloproteinases (MMPs), is increased [25,26]. 

Previous studies have demonstrated the significance of mechanical force in modulating hypertrophic scar (HTS) formation in a mechanically stress-induced murine scar model [27,28]. Sustained mechanical load applied to incisions resulted in fibrotic responses and HTS formation, demonstrating that mechanomodulation is linked to scarring through the stimulation of an inflammatory response [29]. Clinical evidence also supports the importance of mechanical stress on the development of scarring, as the use of a compression device on human patients postoperatively led to reductions in scar size through mechanical offloading in multiple human randomized clinical trials (RCTs) [30,31]. Specifically, a contracting elastomeric silicone dressing device has been shown to significantly improve scars through application of compressive forces to the incision by minimizing impact on the wound through the off-loading of tension [30,32]. Similarly, multiple RCTs of skin taping to improve scar appearance have shown clinical benefit in the appearance of scars [33,34,35,36]. 

In the following paragraphs, we will review the pathways that mediate mechanotransduction and underlie these clinical findings.

## 4. Tensegrity and CTF

The skeletal and membranous components of cells both serve as mediators between physical force and downstream cellular signaling and activity. Tensegrity describes the alignment of structural components of the cytoskeleton in response to mechanical forces in order to preserve the cell’s tensional integrity. The cells are exposed to mechanical prestress as the cytoskeletal components are connected with each other and to the ECM, leading to a continuous balance between opposing forces (e.g., actomyosin contraction is resisted by microtubules intracellularly and by the stiffness of the ECM outside the cell). This ongoing balance of tensional forces provides the cell with stability: when mechanical stress comes from the ECM or surrounding cells, the cytoskeleton’s arrangement changes to counteract this stress. The cytoskeleton is further subdivided into hierarchically subordered structural components that ensure stability themselves, for example the nucleus, the submembranous cytoskeleton, and the actomyosin filament bundles work independently [16,17]. These structural components ensure a connection between the ECM, as well as the cell surface membrane and nuclear chromatin, generating epigenetic modulation of intranuclear programs when the mechanical forces of the surrounding tissue act upon the cell surface [37]. Conversely, cell traction force (CTF) is generated by intracellular actin/myosin interactions that are regulated by α-SMA and TGF-β. Information is then passed on to the ECM via focal adhesions, regulating cell migration and ECM organization and modulating mechanical signal generation [38]. In summary, mechanical stimuli from the outside are transduced to the nucleus and vice versa via tensegrity and CTF, respectively. Tensegrity strongly determines wound healing processes, scar formation and cellular activity, shape, and motility alike [39,40].

## 5. TGF-β

TGF-β represents a crucial soluble factor promoting fibrosis, with TGF-β1 inducing myofibroblast stimulation. The activated myofibroblasts secrete TGF-β, creating a positive feedback loop that further exacerbates fibrosis. As a growth factor, it is involved in structural changes of the cell and the secretion of pro-fibrotic factors, such as α-SMA, which interact with myosin to increase tension [41,42,43]. 

In the ECM, TGF-β1 can bind to the latency-associated peptide (LAP) and the latent TGF-β-binding-protein-1 (LTBP-1). Via integrins, LTBP-1 can be activated by intracellular forces and TGF-β1 is released. As cell tension and force propagation increase, so does TGF-β1 liberation and biomechanical tissue stiffness in fibrosis therefore increases TGF-β1 activation [44]. TGF-β1 itself promotes the expression of genes that are integral to the deposition of the ECM, such as collagens [45], fibronectin [46], and plasminogen activator inhibitor type-1 (PAI-1) [47] and they accumulate in fibrosis.

TGF-β2 is known to have pro-fibrotic properties and accumulation has been found in human fibrotic liver disease [48]. However, TGF-β3 is characterized by the regulation of epidermal and dermal cell motility, which plays a key role in wound repair and promotes wound healing without fibrotic scar formation [49]. In this regard, Ferguson et al. assessed the intradermal administration of avotermin, recombinant, active, human TGF-β3 for scar prevention. Visual assessment of scar formation was performed after six and twelve months on a visual analogue scale (VAS) with the intervention group displaying significant improvement. However, only phase I and II of human clinical trials were passed, whereas phase III not [50]. 

Recently, several studies have investigated pharmacologic therapies targeting TGF-β signaling in fibrotic diseases throughout the body, affecting the kidney [51], lungs [52], and heart, among others [53]. P144, a peptide inhibitor of TGF-β1, was tested by Santiago et al. in mice that received daily injections of bleomycin representing a model of human scleroderma. Their study displayed that there was a significant decrease in skin fibrosis and soluble collagen content after application of P144 lipogel for two weeks. TGF-β and α-SMA levels were lower, resulting in reduced wound scarring and fibrosis [54]. Nevertheless, few of these studies showed positive outcomes in patients [41]. Creating a systemic therapy without impeding otherwise unrelated homeostatic processes remains complicated due to the versatility of TGF-β.

## 6. FAK-ERK-MCP1 Pathway

Focal adhesion kinase (FAK), which is a non-receptor cytoplasmic tyrosine kinase, is one of the key mediators of skin mechanobiology and it is activated after cutaneous injury [27,55]. Mechanical forces potentiate the activation of FAK through phosphorylation following injury of the skin [56,57,58]. FAK contributes to cell signaling through its linking of mechanical stress from the ECM to the cytoplasmic cytoskeleton [59], activating inflammatory pathways. Fibroblasts are recruited to the wound by inflammatory signaling, where their secretion of profibrotic cytokines brings about increased collagen synthesis.

Interestingly, various pathologies that are associated with poor wound healing have been shown to have atypical levels of FAK. Wong et al. demonstrated that mechanical force regulates pathologic scarring through inflammatory FAK-ERK-MCP1 pathways, and that molecular strategies targeting focal adhesion kinase (FAK) can effectively uncouple mechanical force from fibrosis [27] (Figure 2). Thus, FAK remains a potentially promising target for drug approaches to minimize scar formation [27]. Further, FAK is found to be downregulated in diabetes under high glucose conditions as a result of increased calpain 1 activation [60]. Liu et al. have shown that fibroblasts are prone to high glucose stimuli, unlike keratinocytes, which are resistant to high glucose-induced FAK degradation by calpain 1 [60]. Further studies have shown that mice with keratinocyte FAK knockout display delayed wound closure with reduced collagen density and dermal thickness. FAK knockout keratinocytes exhibited hyperactive MMP9 and p38 signaling when cultured in vitro, highlighting FAK as an upstream regulator that is essential to wound healing [61]. The aberrant secretion of MMP9 is a result of blocked FAK in suspended human keratinocytes and the loss of epithelial FAK has been shown to impede normal repair pathways [61]. These findings highlight the significance of properly functioning mechanotransduction pathways during the regeneration processes and that some level of their activity is crucial in proper wound closure.

Pharmacological approaches to locally inhibit FAK have previously shown anti-scarring effects in in vitro and preclinical animal studies [62,63]. In 2018, Ma et al. were able to show that the application of FAK inhibitor (FAKI) pullulan-collagen-based hydrogel scaffolds promotes wound healing with reduced scar formation [64]. FAKI hydrogels inhibited the phosphorylation of FAK in a sustained manner that was mediated by slow release. As a result, collagen deposition and myofibroblast counts were both reduced in FAKI treated wounds. Further, FAKI treated healed skin displayed improved mechanical integrity, as seen by the quantification of the Young’s modulus [64]. Such findings point to the therapeutic effects that targeting mechanotransduction pathways may bring.

## 7. Canonical Wnt Pathway and β-catenin Pathway

Like the FAK-ERK-MCP1 pathway, Wnt/β-catenin signaling has been shown to play an essential role in the self-renewing ability of the skin [65,66]. The canonical Wnt pathway is essential to passing extracellular mechanical signals into the cell through its surface receptors. β-catenin is a structural component of adherens junctions in epithelial cells, regulating cell-cell interactions (Figure 3). After binding to the cytoplasmic domain of E-cadherin, β-catenin binds to α-catenin and it acts as a component of intercellular adhesive junctions [67], which mechanically link cadherins to actin [68]. α-catenin is thought to be essential for linking cadherin to actin filaments.

In response to physical force and the upregulation of Wnt signaling, β-catenin accumulates within the nucleus [69]. Interestingly, fibroblasts expressing increased amounts of β-catenin are found in hypertrophic scars and keloids, implicating β-catenin in cutaneous fibroproliferative diseases [70]. Further, skin biopsies from patients with systemic sclerosis display increased expression of several Wnt molecules, implicating Wnt signaling in fibrotic diseases of the skin [71,72]. As such, the Wnt/β-catenin pathway affects scar formation in the dermis through the upregulation of pro-fibrotic function [73,74], and prolonged activation of Wnt/β-catenin signaling has been observed in human hyperplastic wounds [75]. 

Ray et al. showed that β-catenin provides stability to the epidermis under stress, while the loss of β-catenin results in a loss of response to mechanical stimuli in vitro [76]. The loss of β-catenin weakened tight junctions’ associations with the cytoskeleton and in turn cellular responsiveness to mechanical stress. Increased Wnt signaling was also shown to increase the production of reactive oxygen species (ROS), leading to DNA damage and, thus, increased senescence [77]. Furthermore, the Wnt/β-catenin pathway is connected to TGF-β signaling as β-catenin induces TGF-β to induce fibroblast activity in human skin [78,79]. The manipulation of Wnt/ β-catenin signaling, through both up- and down-regulation, significantly alters the wound healing and scarring responses via the manipulation of mechanosignaling. Regardless, the specific role and changes of Wnt/β-catenin in wounded skin still need to be further elucidated, though manipulation of the pathway could prove to be a promising therapeutic.

## 8. YAP/TAZ

The Hippo pathway is a highly conserved network that moderates tissue growth in adults [80]. Yes-associated protein (YAP) and transcriptional coactivator with PDZ-binding motif (TAZ) are two major downstream regulators that serve as mechanotransducers in response to the stiffness and arrangement of the cytoskeleton [39]. Because the cytoskeleton’s organization and tensional state are influenced by the cell’s physical position in three-dimensional space, mechanical strains are applied to the cell that alter its rigidity and structure [81,82]. While the cell is free from outside stress, YAP/TAZ persists in an inactive state within the cytoplasm that is modulated by mechanical stress upstream. Focal adhesion formation through structures, such as integrin, FAK, and SRC, among others, causes YAP/TAZ to dephosphorylate from their inactive complex and subsequently relocate to the nucleus, where they drive transcription in concert with several coactivators [81,82,83]. Such activity impacts the metabolic function of cells and influences myriad processes throughout the body, from myofibroblast differentiation in fibrosis, to the activity of endothelial cells in angiogenesis [82,84,85]. As such, YAP/TAZ are essential regulators of collective migration and proliferation and they are likely increasingly expressed at sites that are susceptible to higher levels of tension, making them particularly important to scar development [81]. 

The unspecific nature of YAP/TAZ activity and its regulatory role within various profibrotic pathways makes it a promising therapeutic target, although much remains to be elucidated. Accordingly, contemporary research has attempted to identify the essentials of YAP/TAZ signaling in the progression of wound regeneration. Lee et al. showed, in an siRNA-mediated knockdown of YAP/TAZ model, that YAP/TAZ localization to the nucleus is essential for proper wound healing in full-thickness wounds [86]. Another interesting study suggests that YAP functions as a molecular switch of stem/progenitor cell activation in the epidermis and it is essential for epidermal differentiation and proliferation [87,88]. Additionally, YAP is known to influence skin size and tissue overgrowth [89] (Figure 4).

## 9. ILK-PI3K/Akt Pathway

Mechanical strain activates Integrin-linked kinase (ILK)-Phosphoinositide 3-kinase (PI3K)/Akt signaling [90]. Generally, PI3K/Akt activity influences cell proliferation, motility, growth, survival, and apoptosis. There are three classes of PI3Ks, of which Class I is activated by receptor tyrosine kinases (RTKs) or G protein-coupled receptors (GPCRs). Akt is a serine/threonine protein kinase that is activated following recruitment to the plasma membrane and it acts downstream from PI3K [91]. The ILK-PI3K/Akt pathway is activated through transmembrane integrin receptors and it initiates signaling of Akt following PI3K-dependent phosphorylation (Figure 5). Pathway activation in fibroblasts increases their motility, as well as α-SMA expression and collagen I production. Li et al. demonstrated that ILK-overexpression leads to greater scar contracture and scar hypertrophy. Conversely, ILK and PI3K/AKT inhibitors inhibited wound contraction and re-epithelialization, consequently delaying wound healing in vivo. Similar effects were observed following the application of Akt inhibitors in vitro [92].

Moreover, the PI3K/Akt-pathway is a crucial pathway in human keratinocyte differentiation. In fact, the PI3K/Akt-pathway is activated early in the keratinocyte differentiation. Activation is contingent on the expression of epidermal growth factor receptor (EGFR), E-cadherin-mediated adhesion, and Src families of tyrosine kinases, respectively. Blockade of PI3K was shown to cause cell death in keratinocytes, underscoring the importance of balanced PI3K/Akt signaling to proper keratinocyte functionality [93]. 

Investigating the effect of stretch force in vitro, Yano et al. observed that the mechanical stretching of keratinocytes led to the activation of the Akt-pathway, generating proliferative and antiapoptotic signals. Thus, the degradation of keratinocytes is inhibited and skin preservation is promoted [94]. Moreover, previous studies of our lab have shown that Akt activation fluctuates with both the frequency and degree of strain imposed, and that wound tension in mouse models promotes dermal fibroblasts to express the higher activation of Akt [95]. Further, Gao et al. evaluated peaks of PI3K- and p-Akt-expression during reconstruction in the wound healing process and suggested western blot-quantification as a marker for wound time estimation [96]. Akt also serves to activate Mammalian Target of Rapamycine (mTOR), which mediates the fibroblast response to TGF-β [97] and MMP1-inhibition, preventing its proteolytic collagenase activity [98].

The aforementioned studies implicate the ILK-PI3K/Akt pathway as an essential player in wound healing, with both over- and under-expression hindering the regenerative process. As such, the pharmacological modulation of the ILK-PI3K/Akt pathway in fibroblasts and keratinocytes might serve to ameliorate hypertrophic scarring.

## 10. Rho-GTPases

Rho-GTPases are intracellular proteins that mediate between integrins and the cytoskeleton, regulating cell tension, intracellular force, motility, and adherence in response to mechanical force [99]. Rho-GTPases are activated by tension on integrins in focal adhesions as well as by soluble factors, such as growth factors [100] (Figure 6). RhoA, Rac1, and Cdc42 generate contractile forces by coordinating myosin II activity by Rho kinase (ROCK) stimulation [101] and promoting actin filament assembly [102]. As a consequence, wound contracture and fibrosis are increased. Previous studies examining fibroblasts in cardiac and pulmonary fibrosis have determined that the inhibition of Rho-associated signaling reduces collagen synthesis [103]. Bond et al. used Fasudil, a Rho-associated kinase inhibitor, to hinder fibroblast contractility and prevent excessive scarring in rat models. As a result of Fasudil application, wound contracture was reduced when compared to control wounds [104]. Finally, Rho-GTPases represent key mediators for cell proliferation, survival, and motility, not only in fibroblasts, but also in keratinocytes [105]. Hence, future investigations on Rho signaling pathway inhibitors may lead to an effective therapeutic option to prevent fibrosis and contracture in wounds, as well as to treat diseases that are characterized by excessive fibrosis, such as idiopathic pulmonary fibrosis (IPF).

## 11. Calcium-Dependent ion Channels

Mechanosensitive calcium-dependent ion channels serve as integral mechanotransducers, as incidental stretching can force open channels, allowing calcium (Ca^2+^) influx and calcium-dependent pro-fibrotic pathway initiation. The mechanosensitive cation channel TRPV4 (transient receptor potential vanilloid 4) and the voltage-gated L-type channels, among others, influence myofibroblast differentiation through extracellular calcium influx. Active TRPV4 was further shown to enhance actomyosin remodeling and promote nuclear translocation of the α-SMA transcription coactivator, another factor contributing to fibrogenesis [106,107]. 

Calcium levels alter nitric oxide (NO)-pathways that are involved in promoting angiogenesis, proliferation of fibroblasts, epithelial cells, and keratinocytes [108]. 

Ca^2+^-dependent ion channels have proven to be effective therapeutic targets across various diseases. Mukherjee et al. tested several types of calcium channel blockers (CCBs), namely 1 μM of Nifedipine, 1 μM of Verapamil (both L-type blockers), 2.7 μM of Mibefradil (a mixed L-/T-type blocker), 40 μM of NiCl_2_ (selective T-type blocker at this concentration), 30 mM of KCl (depolarizes the cell membrane and inactivates T-type current), and 1 mM of NiCl_2_ (L- and T-type current blocker) in human fibroblasts, showing that intracellular Ca^2+^ oscillations evoked by exogenously applied transforming growth factor-β (TGF-β) were significantly reduced [109]. Such inhibitors are believed to interfere with fibroblast Ca^2+^ influx, disrupting Ca^2+^ oscillations and, in turn, fibroblasts’ abilities to synthesize collagen, thereby protecting from pulmonary fibrosis in humans. Nifedipine was further shown to reduce bleomycin-induced lung fibrosis in rat models, impeding Ca^2+^ influx [109]. The utilization of CCBs has also been demonstrated to encourage wound healing in rat models, with studies assessing the administration of verapamil, diltiazem, difedipine, amlodipine, and azelnidipine. Accelerated wound healing and enhanced tensile strength of the skin were observed in treatment groups when compared with the untreated controls [110,111,112,113].

## 12. GPCRs

G protein-coupled receptors (GPCRs) are characterized by the presence of seven transmembrane domain proteins and they are activated by focal adhesion complexes [114]. GPCRs can promote both pro-fibrotic and anti-fibrotic phenotypes in fibroblasts. Receptor class and downstream signaling pathways determine whether fibrosis is supported or not [115] (Figure 7).

When pro-fibrotic pathways are activated, GPCRs take part in collagen synthesis, as well as fibroblast contraction [116]. Conversely, Roberts et al. explored the ability of GPCRs to inhibit fibrotic development. GPCRs can increase the cAMP levels in human lung fibroblasts, leading to the inhibition of fibroblast proliferation and differentiation. Formoterol, prostaglandin E_2_, treprostinil, and forskolin were shown to reach maximal cAMP responses, whereas partial GPCR-agonists achieved full inhibition of fibroblasts, providing a promising set of targets for novel IPF treatments [117]. Interestingly, it was demonstrated that maximal cAMP response does not correlate with maximal inhibition of proliferation and differentiation in fibroblasts. Moreover, Dooling et al. described the inhibition of tumor fibrosis and decreased force generation of the actomyosin cytoskeleton by the application of GPCR-agonists [118].

## 13. Future Perspectives

In conclusion, mechanotransduction contributes to cellular proliferation, survival, differentiation, mobility, shape, and apoptosis, whether by first-hand effect or downstream paracrine signaling. Thus, both external and internal mechanical forces have the ability to alter phenotypical responses to disease and dysfunction. As the skin serves as the body’s barrier to external stimuli, it is almost always subjected to such mechanical forces that impede wound repair processes and intensify scar production. The development of target molecules to prevent excessive scarring and fibrosis might prove difficult, as target molecules are often characterized by non-specific target cells. Nevertheless, a better understanding and identification of mechanotransduction pathways will contribute to the development of suitable pharmaceutical agents. 

## 14. Conclusions

Scars, keloids, and fibrosis affect a relevant fraction of the population and they often require corrective treatment. However, modern therapies are typically ineffective in the case of scar reduction and patients with large, obvious scars suffer from both a physical and mental perspective. In fact, the exact pathogenesis of many fibrotic diseases, such as keloids, is yet poorly understood, in part because of lacking adequate animal models. The detection of further key players in fibrosis and scar formation is still required [119,120]. Extending beyond experiences of poor wound healing, all surgical procedures inevitably produce a scar, superficially visible or not. From lifesaving cases requiring major incisions to common aesthetic procedures, the resulting scar formation remains an everyday problem in plastic surgery, bringing about a need for improved therapeutics. With an understanding of the factors that contribute to increased scarring and worsened wound healing, such improvements are being made. 

Pathways, including the FAK-ERK-MCP1 and YAP/TAZ, have been shown to impact different stages of regeneration through mechanotransduction, and specific targeting of these pathways has improved scarring. Device approaches have shown promise in mitigating scar formation in humans by countering the scarring effects of stretching with compression, but the ultimate solution will require drug appeal. Understanding mechanotransduction pathways will unravel the mechanisms that regulate scar formation and wound healing and lead to the development of effective therapeutics to reduce scar formation in a number of fibrotic diseases.

## Figures and Tables

**Figure 1 jcm-09-01423-f001:**
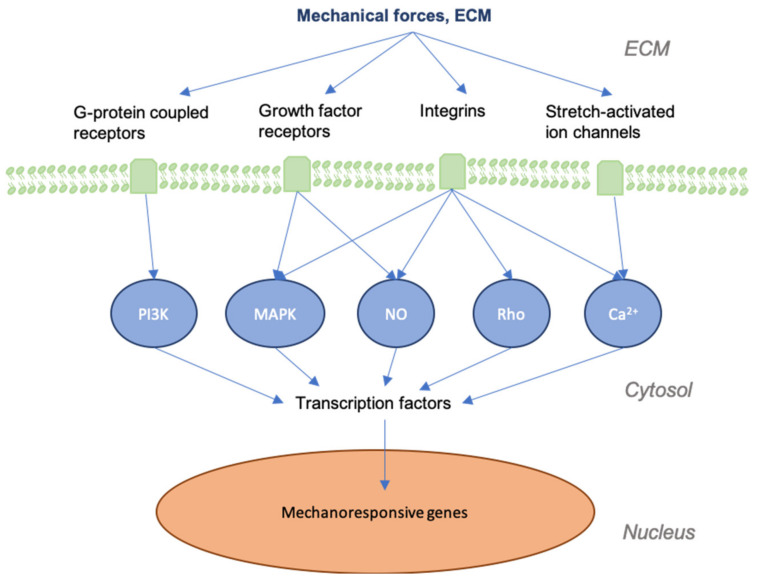
Graphic depicting mechanical forces applied to skin and the associated effects at cellular levels. As skin is stretched, the dermal matrix expands, opening ion channels, receptors, and other mechanotransductors, altering their accessibility to their respective ligands. This alters many of the signaling cascades that are involved in regeneration and the final state of the repaired wound.

**Figure 2 jcm-09-01423-f002:**
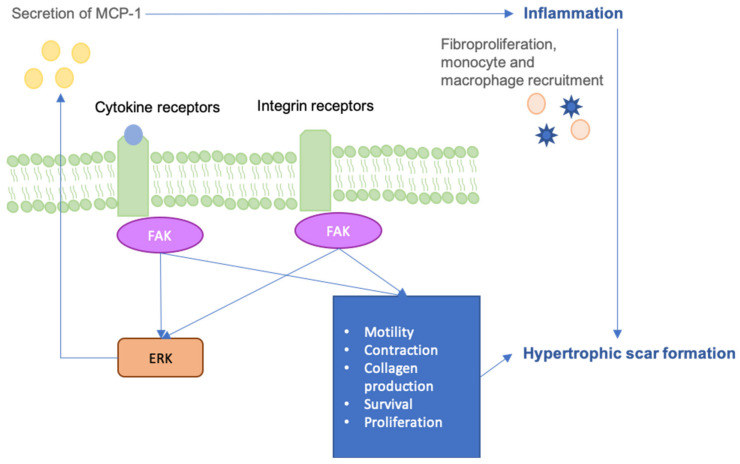
The Focal Adhesion Kinase (FAK) is a critical upstream mediator of these scar-forming processes, linking mechanical stress to inflammatory pathways.

**Figure 3 jcm-09-01423-f003:**
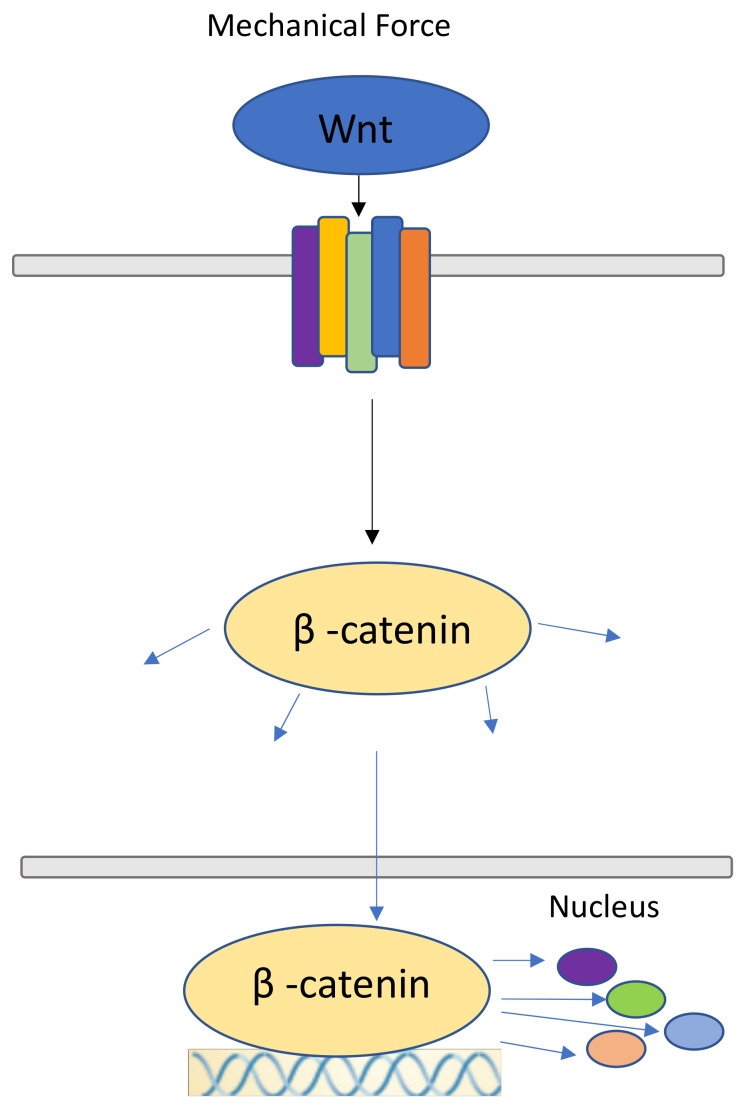
Wnt/ β-catenin signaling pathway. Wnt molecules attached to the respective frizzled-receptors made more prone by mechanical force, releasing β-catenin molecules into the cytoplasm, where they are free to participate in other signaling pathways and can travel to the nucleus. Here, they serve as upstream regulators for other molecules.

**Figure 4 jcm-09-01423-f004:**
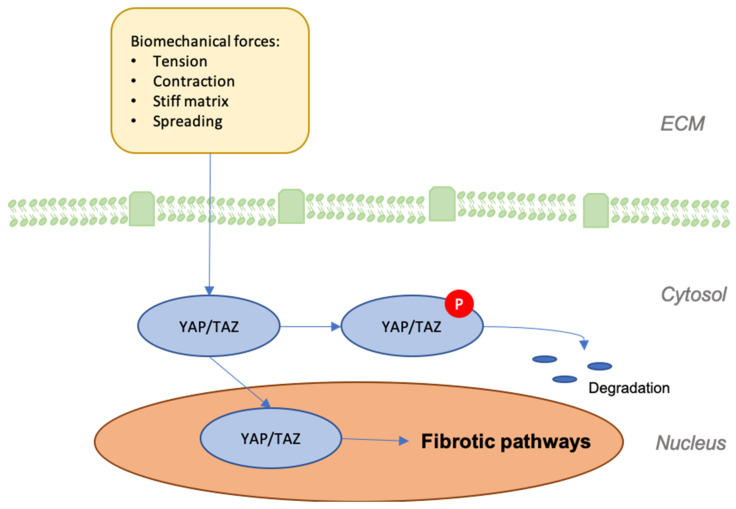
Graphic depicting the effects of mechanical forces on the expression of Yes-associated protein (YAP) and transcriptional coactivator with PDZ-binding motif (TAZ) and their effects on the development of fibrosis.

**Figure 5 jcm-09-01423-f005:**
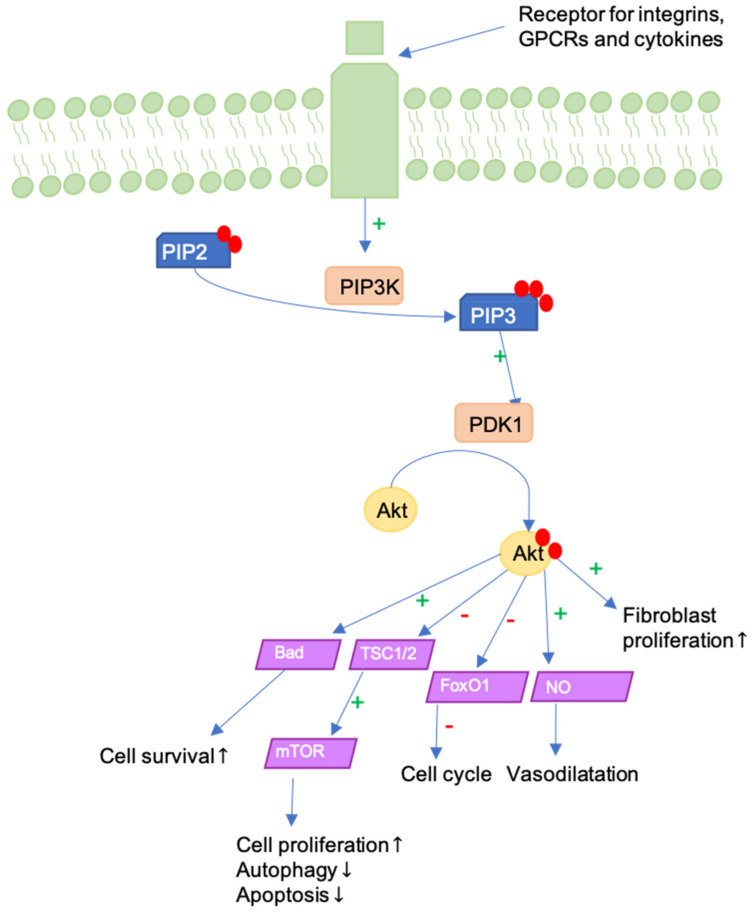
The PI3K/Akt pathway can be activated in different ways, as for example by G-protein coupled receptors (GPCRs), integrin or cytokine receptors and it represents a mediator between ECM and fibroblasts’ cell cycle. In this figure, PI3K-dependent phosphorylation and activation of Akt is represented, which in turn leads to enhanced proliferation and cell survival. Furthermore, apoptosis is regulated, and vasodilatation is promoted.

**Figure 6 jcm-09-01423-f006:**
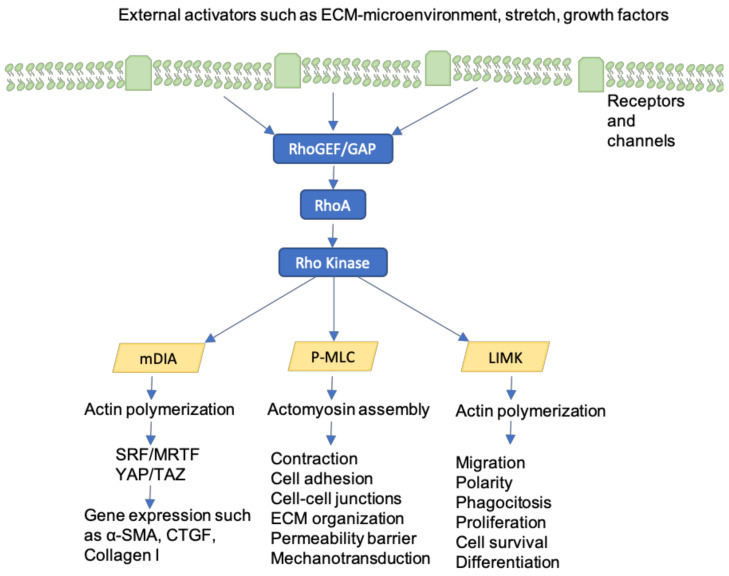
Activation of Rho-GTPases and its’ implications are represented. External factors such as tension on integrins in focal adhesions, cell stretch, growth factors and changes in the ECM environment can lead to Rho-GTPase activation. Through different pathways, cell contraction, motility and consequently wound contracture are promoted. In fact, on the one hand, actin polymerization, α-SMA and collagen I gene expression are enhanced. On the other hand, cell adhesion, cell contraction, ECM organization, migration and polarity are regulated. Finally, cell proliferation, survival and differentiation are coordinated as well.

**Figure 7 jcm-09-01423-f007:**
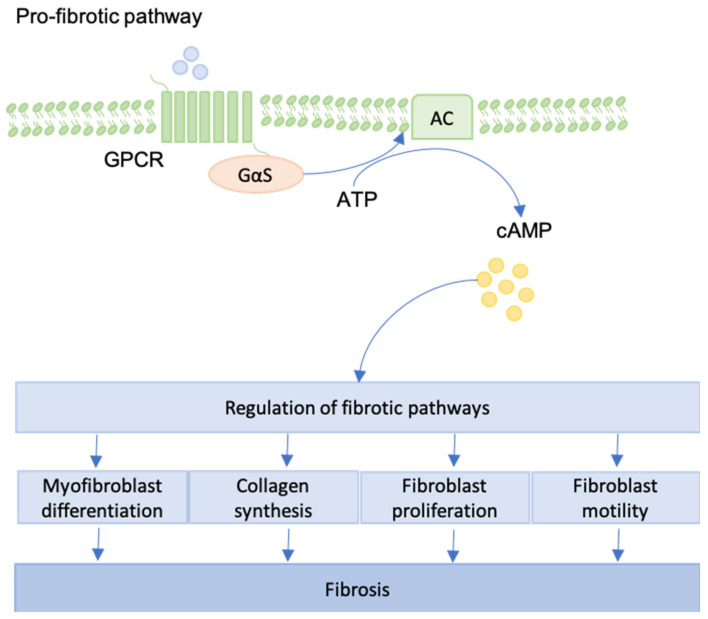
An overview of the pro-fibrotic pathway generated by GPCR-activation is provided. For example, focal adhesion complexes can lead to pro-fibrotic GPCR-activation. Via cAMP generation, fibroblast differentiation, proliferation and motility are regulated. Collagen synthesis can be enhanced as well leading to genesis and progression of fibrosis.

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
