# Peer review of "Mechanotransduction in Wound Healing and Fibrosis"

_jcm, 2020, doi:10.3390/jcm9051423_

Round 1

Reviewer 1 Report

This is a very nice review by Kuehlmann et al., discussing current knowledge on mechanotransduction in fibrosis and wound healing 

Author Response

Thank you very much for your nice feedback. We highly appreciate your kind comments.

Reviewer 2 Report

General: Kuehlmann et all provide us with a well written and complete overview of all important mechanotransduction processes and pathways influencing wound healing and scarring.

3. Similarly, 98 multiple RCTs of skin taping to improve scar appearance have shown clinical benefit in the appearance of scars33 .

Not sure if the best available source is chosen here to support your argument.

4. Only little time is taken to describe the concept/definition of tensegrity, it is hard to follow as it currently stands, maybe worthwhile to extend slightly (1-2 sentences). The use of the concept tensegrity is clear though. 

5. Figure 2: adaptation of figure from Wong et al; the background colouring in your diagrams that are left over from copying are bit distracting. Self-created figures are clear.

6.The roles and relationships of B-catenine and Wnt signalling in the processs of mechanotransduction and scarring is well described and also placed in a broader context than scarring alone.  

8. 'The aforementioned studies implicate the ILK-PI3K/Akt pathway as an essential player in wound 236 healing, with both over- and under-expression hindering the regenerative process. As such, 237 pharmacological modulation of the ILK-PI3K/Akt pathway in fibroblasts and keratinocytes may serve to 238 ameliorate hypertrophic scarring.' 

A more extensive analysis and view on the dilemma (over and under expression both cause impaired regenerative healing) is missing. You imply that there is a range in between where wound healing is not hindered and that this range can be managed by pharmacological modulation. Can you elaborate? 

12. TGF-B is a very significant molecule in the process of scarring. Please consider moving the paragraph to the beginning of your discussion, as now a well-versed reader would continuously wonder when you would discuss its role.

General comment on figures: could be more in the same style. 

Conclusion: Given your thorough overview and comprehensive research I would be interested in hearing your view on what would be further promising avenues for research.

Author Response

We appreciate the constructive comments of the reviewer. In the following, please find our point-by-point reply. Changed parts of the manuscript are highlighted in yellow to provide a better visibility.

General: Kuehlmann et all provide us with a well written and complete overview of all important mechanotransduction processes and pathways influencing wound healing and scarring.

Reviewer comment #1

‘3. Similarly, 98 multiple RCTs of skin taping to improve scar appearance have shown clinical benefit in the appearance of scars33 .’ Not sure if the best available source is chosen here to support your argument.

We agree with the reviewer’s point of view and checked the literature for more argument-supporting studies. We decided to cite a few RCTs showing the efficacy of taping in scar formation. For example, we cited Reiffel et al. who confirmed the benefit of skin taping back in 1995 (1). Also, Rosengren et al. tested the application of adhesive tapes in 63 participants and compared scare appearance with 73 controls after six months. Blinded assessment of overall scar appearance was shown to be significantly better in taped participants (p = 0.004) (2). Moreover, Atkinson et al performed an RCT in which 39 of 70 patients tested paper taping of the scars following cesarean section for 12 weeks. Scars were assessed at 6 weeks, 12 weeks and 6 months. The development of hypertrophic scars was significantly reduced in the intervention group (p = 0.003) (3). These additional sources are found in line 98.

Reviewer comment #2

  1. Only little time is taken to describe the concept/definition of tensegrity, it is hard to follow as it currently stands, maybe worthwhile to extend slightly (1-2 sentences). The use of the concept tensegrity is clear though. 

We agree that it is better to provide the reader with a more detailed description of tensegrity adding some more sentences. We tried to better explain the definition in lines 104 – 115, referring to the detailed studies on the topic performed by Ingber et al (4).

Reviewer comment #3

  1. Figure 2: adaptation of figure from Wong et al; the background colouring in your diagrams that are left over from copying are bit distracting. Self-created figures are clear.

We agree with this comment and in response to this and comment #7, we decided to create the adapted figures 1, 2 and 4 ourselves in order to ensure a homogeneous style. Also, we adapted the figures’ descriptions to our own creations. These changes concern lines 83 – 87, 138 – 140 and 204 – 206.

Reviewer comment #4

6.The roles and relationships of B-catenine and Wnt signalling in the processs of mechanotransduction and scarring is well described and also placed in a broader context than scarring alone.  

We are grateful for the fact that the reviewer appreciated the detailed description of the β-catenine and Wnt signaling pathway and its involvement in the process of scarring.

Reviewer comment #5

  1. 'The aforementioned studies implicate the ILK-PI3K/Akt pathway as an essential player in wound 236 healing, with both over- and under-expression hindering the regenerative process. As such, 237 pharmacological modulation of the ILK-PI3K/Akt pathway in fibroblasts and keratinocytes may serve to 238 ameliorate hypertrophic scarring.' A more extensive analysis and view on the dilemma (over and under expression both cause impaired regenerative healing) is missing. You imply that there is a range in between where wound healing is not hindered and that this range can be managed by pharmacological modulation. Can you elaborate? 

We highly appreciate this feedback but in our opinion elaboration of this specific question will be way beyond the scope of this review.

Reviewer comment #6

  1. TGF-B is a very significant molecule in the process of scarring. Please consider moving the paragraph to the beginning of your discussion, as now a well-versed reader would continuously wonder when you would discuss its role.

Initially, we thought it might keep the reader curious mentioning TGF-β in the end. However, we understand the reviewer’s point of view and it is probably better to move this paragraph at the beginning to keep a hierarchical structure in the importance of the single molecules. We moved the paragraph from lines 317 – 342 to 125 – 150. We also kept the last paragraph of this chapter for a final chapter in the discussion on our view and future perspectives (also regarding comment #8).

Reviewer comment #7

General comment on figures: could be more in the same style. 

In response to this comment and comment #3, we decided to create all figures ourselves and as already mentioned before, these changes are found in lines 83 – 87, 138 – 140 and 204 – 206.

Reviewer comment #8

Conclusion: Given your thorough overview and comprehensive research I would be interested in hearing your view on what would be further promising avenues for research.

We agree that the expression of our view on future perspectives is missing. Thus, We introduced a chapter on future perspectives at the end of the discussion. This chapter is found in lines 356 – 365 (chapter 13).

Reviewer 3 Report

  1. This review is well written and organized, but basically only enumeration of molecules involved in mechanotransduction and fibrosis. Which ones to focus on among these many molecules? Or search new molecules? It would be great if the authors add a paragraph on their original view and expected future research directions based on their updated data.

  1. Chapter 4

There is nothing new, and how it is involved in this topic is not fully clear.

  1. Some of the molecules such as Wnt/β-catenin, Ca channels, and TGF-β have been discussed in this field for a long time. In some of these chapters, it would be better to more highlight what is new as a therapeutic target.

  1. Last paragraph of Chapter 12

This should belong to another chapter.

  1. Chapter 13

It should be discussed that the developmental mechanism of many fibrotic diseases including keloids is still unknown despite the identification of these molecules.

  1. References 45 and 46 are the same.

Author Response

We appreciate the fair and constructive comments of the reviewer. In the following, please find our point-by-point reply. Changed parts of the manuscript are highlighted in yellow to provide a better visibility.

Reviewer comment #1

This review is well written and organized, but basically only enumeration of molecules involved in mechanotransduction and fibrosis. Which ones to focus on among these many molecules? Or search new molecules? It would be great if the authors add a paragraph on their original view and expected future research directions based on their updated data.

We agree that a chapter on our view and future perspectives would be an interesting addition to our review. Therefore, we introduced a new chapter on future perspectives at the end of the discussion. We moved the last paragraph of the chapter on TGF-β to this new chapter and added our opinion on future developments (also with regards to comment #4). This chapter is found in lines 356 – 365 (chapter 13), a more detailed insight in future developments is beyond the scope of this review.

Reviewer comment #2

Chapter 4

There is nothing new, and how it is involved in this topic is not fully clear.

We are thankful for this consideration. We included chapter 4 (Tensegrity and CTF) in our manuscript, because from our point of view it represents an important topic on the mechanical construction of the cell and it explains how external and intrinsic mechanical forces influence cellular stability as well as transcription of genes whose transcripts are molecules that may be involved in cellular motility, contraction, inflammation and fibrosis (4, 5). Thus, wound healing in general, scarring and fibrosis are influenced by tensegrity and CTF (6, 7). In clinical practice, this concept is used in the application of tapes for example, reducing mechanical stress and decreasing the occurrence of hypertrophic scars, providing a better scar appearance (1-3). For the completeness of our work, we wanted to mention this topic shortly. In this regard, we added the studies of Duscher et al. and Noguera et al. as references in line 123.

Reviewer comment #3

Some of the molecules such as Wnt/β-catenin, Ca channels, and TGF-β have been discussed in this field for a long time. In some of these chapters, it would be better to more highlight what is new as a therapeutic target.

We highly appreciate your input. As mechanotransduction in wound healing and fibrosis is a fairly new topic, we tried to sum up in details everything that already exists regarding this topic. Unfortunately, it is not as much as expected with regards to therapeutic targets. Everything we found about therapeutic targets is listed in detail within this review.

Reviewer comment #4

Last paragraph of Chapter 12

This should belong to another chapter.

We agree with this point of view and with the fact that this paragraph should be removed from the TGF-β chapter. Also, we moved the chapter on TGF-β to the beginning of the review (from lines 317 – 342 to 125 – 150). The final paragraph was included in the new chapter on future perspectives instead (lines 355 – 363, see response to comment #1).

Reviewer comment #5

Chapter 13

It should be discussed that the developmental mechanism of many fibrotic diseases including keloids is still unknown despite the identification of these molecules.

We agree with this discussion point and added it to our conclusion, which is now chapter 14, as we added a new chapter before. This issue is discussed in lines 367 – 569: “In fact, the exact pathogenesis of many fibrotic diseases such as keloids is yet poorly understood, in part because of lacking adequate animal models (8, 9). Detection of further key players in fibrosis and scar formation is still required.”

Reviewer comment #6

References 45 and 46 are the same.

We are thankful that the reviewer noticed this inconvenience and we corrected our list of references. Due to the structural changes in the review, this source is number 64 now. In the text it is found in line 184 and 187 and in the list of references it is listed in lines 913 – 915.

References

  1. Reiffel RS. Prevention of hypertrophic scars by long-term paper tape application. Plast Reconstr Surg. 1995;96(7):1715-8.
  2. Rosengren H, Askew DA, Heal C, Buettner PG, Humphreys WO, Semmens LA. Does taping torso scars following dermatologic surgery improve scar appearance? Dermatol Pract Concept. 2013;3(2):75-83.
  3. Atkinson JA, McKenna KT, Barnett AG, McGrath DJ, Rudd M. A randomized, controlled trial to determine the efficacy of paper tape in preventing hypertrophic scar formation in surgical incisions that traverse Langer's skin tension lines. Plast Reconstr Surg. 2005;116(6):1648-56; discussion 57-8.
  4. Ingber DE. Tensegrity I. Cell structure and hierarchical systems biology. J Cell Sci. 2003;116(Pt 7):1157-73.
  5. Wang JH, Lin JS. Cell traction force and measurement methods. Biomech Model Mechanobiol. 2007;6(6):361-71.
  6. Duscher D, Maan ZN, Wong VW, Rennert RC, Januszyk M, Rodrigues M, et al. Mechanotransduction and fibrosis. J Biomech. 2014;47(9):1997-2005.
  7. Noguera R, Nieto OA, Tadeo I, Fariñas F, Alvaro T. Extracellular matrix, biotensegrity and tumor microenvironment. An update and overview. Histol Histopathol. 2012;27(6):693-705.
  8. Chike-Obi CJ, Cole PD, Brissett AE. Keloids: pathogenesis, clinical features, and management. Semin Plast Surg. 2009;23(3):178-84.
  9. Bayat A, Bock O, Mrowietz U, Ollier WE, Ferguson MW. Genetic susceptibility to keloid disease and hypertrophic scarring: transforming growth factor beta1 common polymorphisms and plasma levels. Plast Reconstr Surg. 2003;111(2):535-43; discussion 44-6.

This manuscript is a resubmission of an earlier submission. The following is a list of the peer review reports and author responses from that submission.